# Involvement of HPV Infection in the Release of Macrophage Migration Inhibitory Factor in Head and Neck Squamous Cell Carcinoma

**DOI:** 10.3390/jcm8010075

**Published:** 2019-01-10

**Authors:** Nadège Kindt, Géraldine Descamps, Jérôme R. Lechien, Myriam Remmelink, Jean-Marie Colet, Ruddy Wattiez, Guy Berchem, Fabrice Journe, Sven Saussez

**Affiliations:** 1Department of Human Anatomy and Experimental Oncology, Université de Mons (UMons), Research Institute for Health Sciences and Technology, 7000 Mons, Belgium; nadege.kindt@umons.ac.be (N.K.); geraldine.descamps@umons.ac.be (G.D.); fabrice.journe@umons.ac.be (F.J.); 2Department of Oto-Rhino-Laryngology, Université Libre de Bruxelles (ULB), CHU Saint-Pierre, 1000 Brussels, Belgium; jerome.lechien@umons.ac.be; 3Department of Pathology, Université Libre de Bruxelles (ULB), Erasme Hospital, 1070 Brussels, Belgium; myriam.remmelink@erasme.ulb.ac.be; 4Department of Human Biology & Toxicology, Université de Mons (UMons), Research Institute for Health Sciences and Technology, 7000 Mons, Belgium; jean-marie.colet@umons.ac.be; 5Laboratory of Proteomics and Microbiology, Research Institute for Biosciences, Université de Mons (UMons), 7000 Mons, Belgium; ruddy.wattiez@umons.ac.be; 6Laboratory of Experimental Cancer Research, Luxembourg Institute of Health (LIH), 1526 Luxembourg, Luxembourg; guy.berchem@lih.lu; 7Laboratory of Oncology and Experimental Surgery, Institut Jules Bordet, Université Libre de Bruxelles (ULB), 1000 Brussels, Belgium

**Keywords:** HNSCC, HPV, MIF, 4-IPP, metabolism

## Abstract

Human papilloma virus (HPV) infection has been well-established as a risk factor in head and neck squamous cell carcinoma (HNSCC). The carcinogenic effect of HPV is mainly due to the E6 and E7 oncoproteins, which inhibit the functions of p53 and pRB, respectively. These oncoproteins could also play a role in the Warburg effect, thus favoring tumor immune escape. Here, we demonstrated that the pro-inflammatory cytokine macrophage migration inhibitory factor (MIF) is expressed at higher levels in HPV-negative patients than in HPV-positive patients. However, the secretion of MIF is higher in HPV-positive human HNSCC cell lines, than in HPV-negative cell lines. In-HPV positive cells, the half inhibitory concentration (IC_50_) of MIF inhibitor (4-iodo-6-phenylpyrimidine (4-IPP)) is higher than that in HPV-negative cells. This result was confirmed in vitro and in vivo by the use of murine SCCVII cell lines expressing either E6 or E7, or both E6 and E7. Finally, to examine the mechanism of MIF secretion, we conducted proton nuclear magnetic resonance (^1^H-NMR) experiments, and observed that lactate production is increased in both the intracellular and conditioned media of HPV-positive cells. In conclusion, our data suggest that the stimulation of enzymes participating in the Warburg effect by E6 and E7 oncoproteins increases lactate production and hypoxia inducible factor 1α (HIF-1α) expression, and finally induces MIF secretion.

## 1. Introduction

Head and neck squamous cell carcinomas (HNSCC) are frequent in both men and women worldwide, with approximately 550,000 new cases annually [1]. It has long been recognized that tobacco and alcohol are the two main risk factors for these cancers, but a few years ago, human papilloma virus (HPV) was identified as an additional contributor leading to the development of a new subgroup of HNSCC, mainly associated with HPV-16 and HPV-18 types [2,3]. The impact of HPV infection remains controversial concerning the prognostic values of HNSCC, but recent studies put to light a new blood test that contributes to the detection of circulating tumor HPV DNA (notably HPV-16) and that can predict cancer recurrence [4,5]. It should be noted that differences throughout the world; for example, the prevalence of HPV-16 infection in oropharyngeal cancer, is nearly 60% in North America, whereas it is approximately 30% in Europe (Samples from Germany, United Kingdom and Italy) [6]. However, in the county of Stockholm, researchers have observed an increase of HPV prevalence in tonsil carcinoma by over 90% of cases until 2006 and 2007 [7]. Moreover, a large amount of evidence supports the idea that two HPV-positive cancer subpopulations may be distinguished. The first population is mainly composed of younger adults, primarily Caucasians and nonsmokers, with a favorable prognosis; while the second group includes patients who smoke and use alcohol, and these patients have a poorer prognosis [6,8,9,10].

Macrophage migration inhibitory factor (MIF) is an ubiquitous pro-inflammatory cytokine discovered in 1966 by Bloom and Bennett [11]. This cytokine has been investigated in many clinical and experimental studies in both inflammatory diseases and cancer. Indeed, MIF is involved in cancer progression through different pathways leading to cell proliferation, cell invasion, angiogenesis and tumor immune escape [12]. Implications for MIF in HNSCC have already been reported in several studies, which have established that MIF leads to cancer progression and poorer prognosis, notably in laryngeal carcinoma [13]. Indeed, our previous study has shown that mice that receive a murine squamous carcinoma cells underexpressing MIF and are treated with cisplatin or 5-fluouracil have a better prognosis than mice receiving control cells [13].

The objective of our research is to assess MIF production in HNSCC regarding HPV infection. Indeed, MIF regulation has been studied in several virus-induced cancers, but it has never been examined in HPV-infected carcinomas. Thus, we have evaluated MIF expression in a series of 156 clinical samples including oral cavity and oropharyngeal carcinomas. Moreover, MIF secretion was investigated in vitro and in vivo by using three HPV-negative and three HPV-positive HNSCC human cell lines, as well as murine SCCVII cell lines expressing the E6 and/or E7 HPV oncoproteins. Finally, proton nuclear magnetic resonance (^1^H-NMR) experiments were conducted to examine the mechanism involved in HPV-dependent MIF secretion in HNSCC.

## 2. Materials and Methods

### 2.1. Clinical Characteristics

Formalin-fixed paraffin-embedded oral cavity and oropharyngeal cancer specimens were obtained from 156 patients who underwent curative surgery at Saint-Pierre Hospital (Brussels, Belgium), EpiCURA Baudour Hospital (Baudour, Belgium) and Erasme hospital (Brussels, Belgium) during the years from 1996–2008. The clinical characteristics of the patients are outlined in Table 1 and Table 2, which presents information concerning the patients’ age, gender, tumor localization, tumor histopathological grade, Tumor Node Metastasis (TNM) stage, risk factors, tumor recurrence, and clinical follow-up. This retrospective study was approved by the Institutional Review Board of Jules Bordet Institute (CE2319). All patients included in this study have received information regarding the use of residual human corporal materials for clinical research, and written informed consent was obtained from the patients.

### 2.2. DNA Extraction

The formalin-fixed, paraffin-embedded tissue samples were sectioned, deparaffinized, and digested with proteinase K by overnight incubation at 56 °C. DNA was purified using a QIAamp DNA Mini Kit (Qiagen, Benelux, Belgium), as previously described [14].

### 2.3. Detection of HPV by Polymerase Chain Reaction (PCR) Amplification

HPV DNA detection was performed using PCR with GP5+/GP6+ primers (synthesized by Eurogentec, Liege, Belgium) that amplified a consensus region located within the L1 region of the HPV genome, as previously described [14].

### 2.4. Real-time PCR Amplification of HPV Type-Specific DNA

All DNA extracts were tested for the presence of 18 different HPV genotypes using TaqMan-based real-time quantitative PCR that targeted type-specific sequences of the viral genes, as previously described [14].

### 2.5. RNA Extraction

After five days of cell culture, total RNA was collected in RLT buffer supplemented with β-mercaptoethanol (RNeasy Mini Kit, Qiagen, Venlo, The Netherlands) at 4 °C, and the sample were then centrifuged in RNeasy spin columns. After the wash steps, RNA was collected in RNase-free water, and subjected to DNase treatment as described by the manufacturer. The RNA concentration was evaluated using a NanoDropTM 1000 spectrophotometer (Thermo Scientific, Wilmington, DE, USA). RNA quality was assessed, based on the RNA profile generated by a Bioanalyzer 2100 (Agilent Technologies, Santa Clara, CA, USA).

### 2.6. Real-Time PCR for MIF mRNA Quantification

MIF mRNA expression was quantified by real-time PCR. Complementary DNA (cDNA) was synthesized using a standard reverse transcription method (qScript cDNA SuperMix, Quanta Biosciences, Gaithersburg, MD, USA). Real-time PCR reactions were performed using the SYBR Green PCR Master Mix (Applied Biosystems, Foster City, CA, USA) and sequence-specific primer sets designed with PrimerBank [15] for MIF (forward: 5′-GCAGAACCGCTCCTACAGCA-3′, reverse: 5′-GGCTCTTAGGCGAAGGTGGA-3) and for 18S (forward: 5′-GCGGCGGAAAATAGCCTTTG-3′, reverse: 5′-GATCACACGTTCCACCTCATC-3′) (Life Technologies, Ghent, Belgium). The amplification was performed on a LightCycler 480 System (Roche Diagnostics GmbH, Mannheim, Germany) using an initial activation step (95 °C for 10 min) followed by 40 cycles of amplification (95 °C for 15 s and 60 °C for 60 s). Melting curves from 60 °C to 99 °C were assessed to evaluate PCR specificity. A preliminary analysis demonstrated linear and similar amplification efficacies. Relative quantification was determined by normalizing the cycle threshold (CT) of MIF with the CT of β-actin (loading control) using the 2−ΔCT method.

### 2.7. Immunohistochemistry of p16

Each HPV-positive case was further immunohistochemically evaluated for p16 expression, using the recommended mouse monoclonal antibody (CINtec p16, Ventana, Tucson, USA) [16], and an automated immunostaining protocol (Bond-Max, Leica Microsystems, Wetzlar, Germany). Immunohistochemistry was performed as previously described [9].

### 2.8. Immunohistochemistry for MIF Protein Detection

MIF expression was detected by immunohistochemistry in tumor samples, and was performed on paraffin-embedded sections, as previously detailed [17].

### 2.9. Computer-Assisted Morphometry

A morphological examination of immunostained tissue sections was carried out on a Zeiss Axioplan microscope equipped with a color change-couple device (CCD) camera (ProgRes C10plus, Jenoptik, Jena, Germany). Morphometric analysis was performed using KS 400 imaging software (Carl Zeiss Vision, Hallbergmoos, Germany), as described previously [13].

### 2.10. LC MS/MS Analysis and Data Processing

Protein identification and quantification were performed using a label-free strategy on an UHPLC-HRMS platform (Eksigent 2D Ultra and AB SCIEX TripleTOF 5600). The peptides (2 μg) were separated on a 25 cm C18 column (Acclaim PepMap100, 3 μm, Dionex, Thermo Scientific, Wilmington, DE, USA) using a linear gradient (5–35% over 120 min) of acetonitrile (ACN) in water containing 0.1% formic acid at a flow rate of 300 nL min^−1^. To obtain the highest possible retention time stability, which is required for label-free quantification, the column was equilibrated with a 10× volume of 5% ACN before each injection. Mass spectra (MS) were acquired across a 400–1500 m/z range in high-resolution mode with a 500 ms accumulation time. The precursor selection parameters were as follows: 200 cps intensity threshold, 50 precursor maximum per cycle, 50 ms accumulation time, and 15 s exclusion after one spectrum. These parameters led to a duty cycle of 3 s per cycle, ensuring that high-quality extracted ion chromatograms (XICs) were obtained for peptide quantification.

ProteinPilot Software (v4.1) was used to conduct a database search against the UniProt Trembl database (09/30/2011 version), which was restricted to Homo sapiens entries. The search parameters included differential amino acid mass shifts for carbamidomethyl cysteine, all biological modifications, amino acid substitutions, and missed trypsin cleavage.

For peptide quantification, PeakView was used to construct XICs for the top five peptides of each protein identified with a false discovery rate (FDR) of lower than 1%. Only unmodified and unshared peptides were used for quantification. Peptides were also excluded if their identification confidence as determined by ProteinPilot was below 0.99. A retention time window of 2 min, and a mass tolerance of 0.015 m/z were used. The calculated XICs were exported into MarkerView, and signals were normalized based on the summed area of the entire run. Only proteins presenting a fold-change of greater than/less than 1.5/0.6 with a p-value lower than 0.05 across the three biological replicates analyzed were accounted for in the metabolic characterization. Fold-changes were assessed by using a Student’s t-test. Finally, proteins identified by one peptide were validated manually.

### 2.11. Cell Culture

UPCI-SCC-131, Detroit 562, UPCI-SCC-90, and UPCI-SCC-154 (DSMZ, Braunschweig, Germany) cell lines were grown in minimum essential medium (MEM, Gibco Life Technologies, Paisley, UK) supplemented with 10% fetal bovine serum, 2 mM l-glutamine, 1% penicillin/streptomycin, and 1% nonessential amino acids (Gibco Life Technologies, Paisley, UK). The FaDU and 93VU-147T cell lines were grown in Dulbecco’s modified Eagle medium (DMEM, Lonza, Verviers, Belgium) supplemented with 10% fetal bovine serum (FBS), 2% l-glutamine, and 1% penicillin/streptomycin. The 93VU-174T cell line was obtained from Dr. de Winter (University Medical Center of Amsterdam, Amsterdam, The Netherlands).

Mouse SCCVII cells were transfected with three different vectors, to express either the HPV oncoproteins E6 or E7, or both E6 and E7 in the Radiation Oncology Department at the Université Catholique de Louvain (Prof. Vincent Grégoire), as previously described [18]. Cells were cultured in T-flasks containing minimum essential medium (MEM) supplemented with 10% FBS, 2 mM L-glutamine, 1% antibiotic/antimycotic mix, and 1% nonessential amino acids (Gibco Life Technologies, Paisley, UK).

A MIF knockdown (MIF-KD) cell line derived from SCCVII, as well as a matched control with normal MIF expression, was generated in the Department of Pediatrics at the Medical College of Wisconsin (Dr. Bryon Johnson). Knock-down of MIF expression in the SCCVII cell line was achieved by using the BLOCK-iT Lentiviral RNAi from Invitrogen (Thermo Scientific, Carlsbad, CA, USA), following a procedure detailed in a previous publication [19]. Cells were cultured as described previously [13].

Routine cell culture was carried out at 37 °C in a humidified cell incubator at 5% CO_2_.

For the hypoxic condition, human HPV-negative and HPV-positive cell lines were maintained in a hypoxia chamber (InvivO2 400 Hypoxia Workstation, Ruskinn) in a humidified atmosphere containing 5% CO_2_ and 0.1% O_2_ at 37 °C for 48 h.

### 2.12. MIF Inhibitor

The MIF inhibitor 4-iodo-6-phenylpyrimidine (4-IPP) was purchased from Tocris Bioscience (Bristol, UK). Stock solutions of the compound were prepared in ethanol, stored at −20 °C, and used within one month.

### 2.13. Determination of MIF Concentrations

Concentrations of MIF in serum and conditioned medium were assayed by a sandwich enzyme-linked immunosorbent assay (ELISA) using a commercial kit (DuoSet ELISA Development kit, R&D Systems, Minneapolis, MN, USA). The assays were carried out according to the instructions provided by the supplier. MIF concentrations in samples of serum and conditioned medium were determined by interpolation from a reference curve established from increasing concentrations of recombinant human or mouse MIF.

### 2.14. IC50 Determination for the MIF Inhibitor by a Crystal Violet Stain Assay

Cells were seeded in 96-well plates. The following day, cells were fed fresh medium (DMEM or MEM, 10% FBS) with or without 4-IPP. After a 3-day exposure, the culture medium was discarded, and cells were fixed with 1% glutaraldehyde. Following fixation, cells were stained with 1% crystal violet. Destaining was performed under gently running tap water, and cell monolayers were lysed in 0.2% Triton X-100. The absorbance of cell lysates was measured at 570 nm using a LabSystems Multiskan MS microplate reader (ThermoScientific, Pittsburgh, PA, USA).

### 2.15. Western Blotting for the Evaluation of HIF-1α and MIF Expression

Human HPV-positive and HPV-negative cell lines and SCCVII CT, E6, E7, and E6/E7 murine cell lines were plated in Petri dishes, cultured for five days, and lysed using detergent solution (RIPA buffer) supplemented with protease and phosphatase inhibitors (All reagents from Pierce, Thermo Scientific, Pierce, Washington, USA). Protein concentrations were determined by a BCA protein assay (Pierce) using bovine serum albumin as the standard. Extracted proteins (40 μg) were loaded on 4–20% Mini-PROTEAN TGX gels (SDS) (Bio-Rad Laboratories, München, Germany) and were electrotransferred onto nitrocellulose membranes (iBlot® Dry Blotting System, Life Technologies-Invitrogen, Ghent, Belgium). Immunodetection was performed using an anti-HIF-1α antibody (1/600), an anti-MIF antibody (1/500), and an anti-actin antibody (1/500) (Pierce). A peroxidase-labeled anti-rabbit IgG antibody (1/5000) (Amersham Pharmacia Biotech, Roosendaal, The Netherlands) was used as the secondary antibody. Bound peroxidase activity was detected using the SuperSignal® West Pico Chemiluminescent Substrate (Pierce) following the manufacturer’s instructions. The bands were visualized by exposing the membranes to photosensitive film (Hyperfilm ECL, Amersham Pharmacia Biotech). Bands intensities were quantified using ImageJTM software (a public domain image software developed by W. Rasband at the Research Services Branch of the National Institute of Mental Health, NIH, Bethesda, MD, USA).

### 2.16. Animal Study

Experiments were conducted on 40 female C3H/HeN mice (Charles River Laboratory, L’Arbresle, France). The animals were maintained and handled in compliance with the guidelines issued by the Belgian Ministry of Trade and Agriculture. SCCVII cells transfected with vectors expressing E6/E7 or with control vector (CT) were inoculated in the mylohyoid muscle following a procedure detailed in a previous publication [18]. Animals were checked for tumor onset, and were euthanized when they exhibited either a tumor diameter exceeding 15 mm, or a weight loss of more than 20%. Postmortem, blood samples were collected and processed for serum.

### 2.17. Cell Sample Preparation for NMR Analysis

Cells of each human cell line were seeded in six 150 mm cell culture dishes, and were maintained in complete MEM. After four days, the medium was replaced by MEM without FBS. After 48 h, the medium was harvested, centrifuged for 10 min at 1200 rpm and stored at −80 °C until NMR analysis. Then, the cells were washed twice with ice-cold Dubelcco’s Phosphate Buffer Saline (DPBS). Then the cells were quenched using 3 mL of cold methanol, detached using a cell scraper and pipetted into a 15 mL centrifuge tube to which 702 µL of ice-cold distilled water was added. All cell extracts were vortexed and suspended in a solution of chloroform/methanol/water in a ratio of 4:3:2. After centrifugation for 30 min at 15000 g, the upper aqueous layer was extracted and used for further analysis; the organic phase was discarded. The aqueous phases from the samples were dried at 25 °C using a centrifugal vacuum concentrator (Eppendorf) and stored at −80 °C until NMR analysis.

### 2.18. H-NMR Sample Preparation and Spectroscopy

A total of 700 µL and 250 µL of phosphate buffer (0.04M NaH2PO4 and 0.2M Na2HPO4, pH 7.4) was added to dried samples and to 500 µL of media, respectively. Samples were then centrifuged for 10 min at 13,000 g, and 650 µL of each sample was transferred to a 5 mm NMR tube, to which 50 µL of 3.5 mM 3-(trimethylsilyl)-propionic-2,2,3,3-d4 acid (TSP) prepared in 100% D2O, was added. The samples were analyzed on a Bruker Avance 500 spectrometer (11.8 T) operating at 500 MHz for proton observation with a 5 mm BBI 1H/D-probe. A one-dimensional spectrum was acquired at 297 °K, using a 1D-NOESY PRESAT pulse sequence. For each samples, 256 free induction decays (FID) were collected using a spectral width of 10,330.578 Hz, an acquisition time of 2.65 s, and a pulse recycle delay of 3 s. After 1H-NMR acquisition, the FID signal was imported into MestReNova 10.0 software (Mestrelab Research, Santiago de Compostela, Spain) for Fourier transformation. Then, the spectra were automatically phased and baseline-corrected, and calibrated against TSP. The resonance of the methyl groups in TSP was arbitrarily set to 0.00 ppm. The spectral region from 0.08–10.00 ppm was automatically reduced to 496 integrated regions (bins) of 0.02 ppm width each. The regions from 4.50–5.20 ppm containing the residual water signal were removed. Each integrated region was normalized to the total spectral area.

The integrated reduced data were imported into SIMCA-P+12 (Umetrics, Umea, Sweden) for PLS-DA. For the quantification of lactate, the resonance with the higher resolution was selected and fitted. An area under the curve (AUC) in arbitrary units was obtained (MestReNova software). This method allowed for statistical comparison between the cell lines.

### 2.19. Statistical Analyses

SigmaPlot^®^ 11 software was used for statistical analyses. Parametric analyses were conducted with the Student’s t-test or with analysis of variance for comparison among more than two groups, and pairwise comparisons were performed using the Holm–Sidak method. Nonparametric analyses were performed with the Mann-Whitney U test (two groups). A *p* ≤ 0.05 was considered to indicate a statistically significant difference.

## 3. Results

### 3.1. Tissue MIF Expression is Decreased in the Oral Cavity and in Oropharyngeal Carcinomas Infected with HPV

The immunohistochemical staining of MIF was examined in two series of 117 cases of oral cavity, and 39 cases of oropharyngeal carcinomas. We did not find statistical correlations between MIF expression and age, gender, tumor localization, histological grade, tumor stage or alcohol and tobacco consumption in these series (Table 1 and Table 2). However, a staining intensity analysis demonstrated that oropharyngeal and oral cavity cancer tissues infected with transcriptionally active HPV (p16+) showed a decrease in MIF expression compared to oropharyngeal and oral cavity cancer tissues not infected by HPV (n = 21 and n = 65 respectively) (Figure 1, *p* = 0.001 and *p* = 0.004 respectively, Kruskal-Wallis test).

This result was confirmed by a previous proteomic analysis comparing HPV+ve versus HPV−ve tumors, which indicated that the MIF expression was two-fold lower in an HPV+ve oral cavity cancer tissue as compared to a HPV−ve oral cavity cancer tissue (*p* = 0.016, Student’s *t*-test, data not shown).

### 3.2. MIF mRNA Synthesis and Protein Secretion is Increased in HPV-Infected Human Head and Neck Cancer Cell Lines

To evaluate the differential expression of MIF in HNSCC cells with regard to HPV status, we compared the MIF messenger RNA (mRNA) expression in three human HPV-negative (FaDu, UPCI-SCC-131 and Detroit 562) and three HPV-positive (UPCI-SCC090, UPCI-SCC154 and 93VU147T) cell lines. The quantitative analysis of the MIF mRNA expression by real-time PCR demonstrated that the MIF mRNA expression was three times greater in HPV-positive cell lines than in HPV-negative ones (Figure 2A, *p* <0.001, Student’s *t*-test), while the MIF protein expression was equally the same among all of the cell lines (Figure 2B).

Moreover, the concentration of MIF was examined in the culture supernatants of the HNSCC cell lines and the SCCVII MIF Knockdown (KD) and Scramble (sc). The conditioned medium from HPV-positive cell lines showed a significantly higher MIF level than the medium originating from HPV-negative cell lines (Figure 2C, *p* = 0.04, Student’s *t*-test). As expected, the conditioned medium from SCCVII MIFKD exhibited a significantly lesser MIF level than the medium from SCCVII MIFsc cells (Figure 2E, *p* = 0.045, Student’s *t*-test).

Finally, the use of 4-IPP, an inhibitor of MIF, in HPV-negative and HPV-positive HNSCC cell lines revealed that the concentrations required to achieve a 50% inhibition of cell proliferation were significantly higher in HPV-positive cell lines than in HPV-negative cell lines (Figure 2D, *p* = 0.018, Student’s *t*-test). This result agrees with the observations that MIF was more highly secreted by HPV-positive cell lines and that higher concentrations of 4-IPP were required to block this higher level of MIF, and consequently to inhibit cell proliferation.

### 3.3. MIF Secretion is Increased in Murine Head and Neck Cancer Cell Lines Transfected with E6/E7 Oncoproteins In Vitro and In Vivo

To investigate the role of HPV oncoproteins in MIF secretion, we analyzed the concentrations of MIF in the cell culture medium of SCCVII CT, E6, E7, and E6/E7. This analysis showed that cells expressing HPV oncoproteins released more MIF than the control cells (Figure 3A, *p* <0.001, one-way-analysis of variance (ANOVA) test). Furthermore, the murine SCCVII cell lines expressing the HPV oncoproteins E6 and/or E7 were exposed to 4-IPP for three days, to examine the resistance to 4-IPP, as measured by cell proliferation. The results showed that SCCVII E6, E7, and E6/E7 cells were all more resistant to the MIF inhibitor compared to the SCCVII CT cells, further demonstrating that E6 and E7 were involved in MIF secretion (Figure 3B, *p* <0.001, one-way-ANOVA test). Finally, the in vitro data were validated using an orthotopic animal model to confirm that MIF was more highly secreted by cells expressing HPV oncoproteins. SCCVII E6/E7 or SCCVII CT cells were injected into the floor of the mouth of C3/Hen mice. As expected from in vitro data, mice receiving SCCVII E6/E7 cells presented higher serum MIF levels than mice receiving SCCVII CT cells (Figure 3C, *p* = 0.013, Mann-Whitney test).

### 3.4. Lactate Production and HIF-1α Expression are Increased in HPV-Positive Cell Lines

We hypothesized that MIF secretion may be affected by a Warburg effect involving E6 and E7 oncoproteins, as observed in lung cancer [20]. To assess this hypothesis, a nuclear magnetic resonance analysis was performed; the partial least-squares discriminant analysis (PLS-DA) allowed us to divide the HPV-positive cells from the HPV-negative ones (Figure 4A, CV-ANOVA, *p* <0.001). The loading plot showed that only lactate comes through the analysis (data not shown). Thus, we examined the lactate production in both the intracellular and extracellular compartments of HPV-negative and HPV-positive cells. The results displayed a significant increase in lactate production in the culture medium (Figure 4B, *p* <0.001, Mann–Whitney test) as well as in the intracellular compartment (Figure 4C, *p* = 0.024, Mann-Whitney test) of the HPV-positive cells compared to the lactate production in the culture medium and the intracellular compartment of the HPV-negative cells.

Besides, it was reported that the induction of the Warburg effect by the HPV oncoproteins through the mTOR signaling pathway leads to the accumulation of HIF-1α [21]. To verify whether HPV oncoproteins play a role in the increase in HIF-1α in HNSCC, we analyzed HIF-1α expression by Western blotting in HPV-positive and HPV-negative human cell lines, as well as in SCCVII CT, E6, E7, and E6/E7 murine cell lines. The data in Figure 5A demonstrate that globally HPV-positive human cell lines and the SCCVII E6/E7 murine cell line expressed more HIF-1α than the HPV-negative human cell lines and the SCCVII CT murine cell line, respectively, under normoxic culture conditions. Additionally, under hypoxic conditions, which stimulate HIF-1α expression and activity, HPV-negative cell lines (FaDu, Detroit 562 and UPCI-SCC131) released more MIF in the culture medium (Figure 5B, *p* <0.001, Student’s *t*-test) than did the HPV-positive ones, thus demonstrating that the increase in HIF-1α led to the rise in MIF secretion. However, the data in Figure 5C show that, under hypoxia, HPV-positive cell lines (UPCI-SCC154, UPCI-SCC090 and 93VU147T) did not secrete significantly more MIF than they did under normoxia; this effect was potentially due to a high baseline level of HIF-1α expression and MIF secretion in these cells (Figure 5C, *p* = N.S., Student’s *t*-test).

## 4. Discussion

As we have previously seen in oral carcinoma, there is no correlation between MIF expression and age, gender, tumor localization, histological grade, tumor stage, or alcohol and tobacco consumption [22]. So, MIF expression does not seem to correlate with these clinical characteristics. However, our data reports an important relationship between MIF secretion and HPV infection, an additional HNSCC risk factor.

HPV infection occurs more often in the oropharyngeal region of the head and neck, and a high proportion of infection are caused by HPV-16 type [6]. The oncoproteins E6 and E7 play a crucial role in the neoplastic transformation of host cells by inactivating p53 and pRb, respectively [23]. In this study, we have demonstrated that, under normoxic conditions, the E6/E7 oncoproteins enhance expression of HIF-1α in HNSCC cells, as it was also reported for human cervical carcinoma cells [24]. Our investigation also showed that HPV infection is conducive to an acidic environment, as evidenced by the increase in lactate production, both in the intracellular compartment and in the culture medium. In addition, we have observed, for the first time, that MIF expression decreased in the tumor cells of HNSCC patients with HPV infection, and that MIF secretion was increased in HPV-positive cell lines, an observation that we confirmed by demonstrating resistance to an inhibitor of MIF (4-IPP) in human cell lines that were infected with HPV, as well as in murine cell lines expressing the E6 and/or E7 oncoproteins. These data were further supported by the increase in the serum MIF in mice receiving the SCCVII E6/E7 cells. In view of these findings, we hypothesized that the acidic environment induced by HPV explains the increase in MIF secretion through the activation of HIF-1α. Consequently, the MIF mRNA level increases, as we have reported in human HPV-positive cell lines, thus leading to elevated protein synthesis and the secretion of MIF.

Other researchers have demonstrated that the E6 and E7 oncoproteins also take part in a perturbation of cellular metabolism termed the Warburg effect, which is a hallmark of cancer cells. Indeed, it was demonstrated that E6 interacts with the transcription factor HIF-1α to induce glycolysis under hypoxia [25]. Moreover, previous studies have shown that the E6 oncoprotein promotes the activity of the mammalian target of rapamycin (mTOR) signaling pathway, thus leading to the accumulation of HIF-1α, pyruvate kinase, lactate dehydrogenase, and pyruvate dehydrogenase kinase 1 [21]. The accumulation of these enzymes induces the Warburg effect by increasing the rate of glycolysis and the production of lactate. It was also reported that the E7 oncoprotein induces the acetylation of the pyruvate kinase M2 isoform (PKM2), thus promoting the dimeric form of PKM2, which leads to the conversion of pyruvate to lactate through lactate dehydrogenase (LDHA) [26].

Furthermore, the high-risk HPV E2 protein, which facilitates the integration of the HPV genome into the host cell genome, could also interact with mitochondrial membranes, thus leading to the production of reactive oxygen species (ROS). This ROS generation correlates with HIF-1α upregulation and lactate production [27]. In contrast, keratinocytes infected with HPV-16 exhibit an increase in monocarboxylate transporter 4 (MCT-4) [28], which is known to participate in lactate export from stromal cells, releasing lactate into the microenvironment and favoring a reverse-Warburg effect for cancer cells. All of these observations established that HPV infection leads to an acidification of the microenvironment. These phenomena may be illustrated by the observations that MIF secretion increased after ROS production, as was demonstrated in clear cell renal cell carcinoma, breast cancer, and lung cancer [29]. Additionally, it is well-known that HIF-1α increases the expression of MIF. Conversely, MIF protects HIF-1α from proteasomal degradation in cancer cells [30,31,32]. However, the mechanism of MIF secretion remains unknown, and it needs further investigation.

The production of lactate and MIF by cancer cells infected with HPV could partly explain why such cells are poorly immunogenic. Indeed, some studies have demonstrated that HPV-positive tumors contain a low number of tumor-associated immune cells, such as Langerhans cells and cytotoxic T cells, while they recruit high levels of regulatory T lymphocytes [18,33,34,35]. In addition, lactate production induces the recruitment of immunosuppressive cells such as tumor-associated macrophages (TAMs) and myeloid-derived suppressor cells (MDSCs) [36]. Moreover, it was demonstrated that lactate inhibits the migration of CD4+ T lymphocytes [37]. Concerning MIF, it was observed that the increase in MIF expression induced a reduction in the number of CD3+ T cells in the stromal compartment of laryngeal carcinoma tissues [13]. Furthermore, Dumitru et al. demonstrated that MIF promotes tumor-associated neutrophil recruitment in a CXCR2-dependent manner in vitro; this recruitment enhances the migratory properties of HNSCC tumor cells [38]. Finally, MIF contributes to tumor immune escape by inhibiting CD8+ T lymphocyte and natural killer cell responses through the downregulation of the NKG2D receptor in ovarian cancer cells [39].

In conclusion, we have reported that MIF is more highly secreted in HPV-positive cells than in HPV-negative cell lines, probably by a specific still-unknown mechanism of MIF secretion, and that the E6 and E7 oncoproteins are implicated in this effect. Based on our data and literature, we explain this relationship by the fact that HPV oncoproteins interact with several pathways and metabolic processes, notably, the Warburg effect, thus generating MIF production. Figure 6 summarizes these pathways and our hypothesis. First, E6 activates mTOR, which leads to the accumulation of HIF-1α that induces the expression of several glycolytic enzymes such as PKM2, LDHA, and the lactate transporter MCT4; in addition, HIF-1α induces MIF expression [21,32,40]. Second, the E7 oncoprotein not only promotes the PKM2-mediated generation of lactate, but also induces HIF-1α expression, thus leading to MIF expression [24,26]. Therefore, the increase in MIF secretion as an immune system repressor, and the production of lactate through the Warburg effect, both by HPV oncoproteins, may lead to the development of a pro-tumoral microenvironment.

## 5. Conclusions

To conclude, we hypothesized that lactate production and HIF-1α expression induces MIF secretion via the HPV E6 and E7 oncoproteins, by the stimulation of enzymes participating in the Warburg effect. This production could lead to development of a pro-tumoral microenvironment.

## Figures and Tables

**Figure 1 jcm-08-00075-f001:**
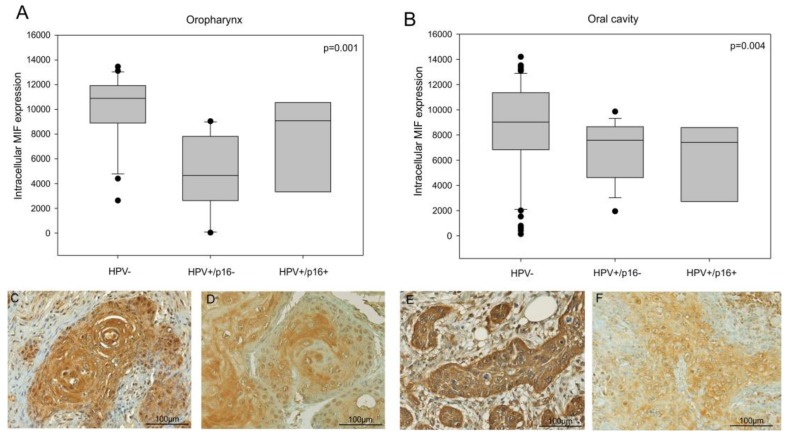
Intracellular migration inhibitory factor (MIF) expression in head and neck cancer patients. (**A**) Quantitative analysis of MIF expression in a series of 39 oropharyngeal cancer patients, including 21 Human Papilloma Virus negative (HPV-ve) cases and 18 HPV+ve cases (*p* = 0.001, Kruskal–Wallis test) and (**B**) 117 oral cavity cancer patients, including 65 HPV-ve cases and 52 HPV+ve cases (*p* = 0.004, Kruskal–Wallis test). (**C**,**D**) Immunohistochemistry of MIF in HPV-ve (**C**) and HPV+ve (**D**) oropharyngeal cancer cases and (**E**,**F**) in HPV-ve (**E**) and HPV+ve (**F**) oral cavity cancer cases.

**Figure 2 jcm-08-00075-f002:**
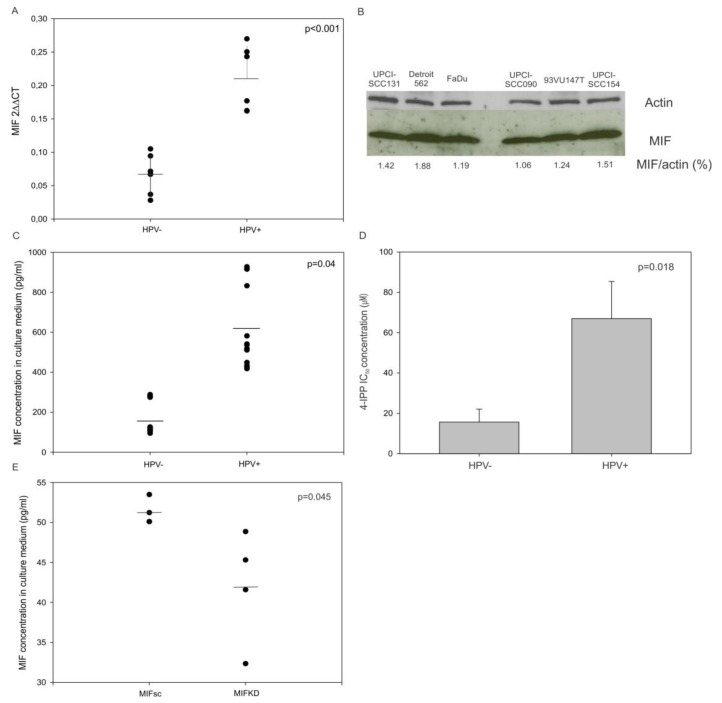
Relative MIF messenger RNA (mRNA), protein expression and macrophage migration inhibitory factor (MIF) concentration in culture medium from mouse and human culture cell lines and the response to MIF inhibitor. (**A**) Upregulation of MIF mRNA in human papilloma virus (HPV)-positive cell lines compared to the MIF expression in HPV-negative cell lines (*p* <0.001, Student’s *t*-test). (**B**) Western blot analysis showing the MIF expression in HPV-ve and +ve cell lines. (**C**) Increased MIF level in the culture medium of HPV+ve cells (n = 3) compared to the MIF level in the culture medium of HPV-ve cells (n = 3) (*p* = 0.04, Student’s *t*-test). (**D**) Concentration of 4-iodo-6-phenylpyrimidine (4-IPP) required to achieve 50% inhibition of cell proliferation in human HPV-ve and HPV+ve cell lines (*p* = 0.018, Student’s *t*-test). (**E**) Increased MIF levels in the culture medium of SCCVII MIFKD cells (n = 4) compared to the SCCVII MIFsc cells (n = 3) (*p* = 0.045, Student’s *t*-test).

**Figure 3 jcm-08-00075-f003:**
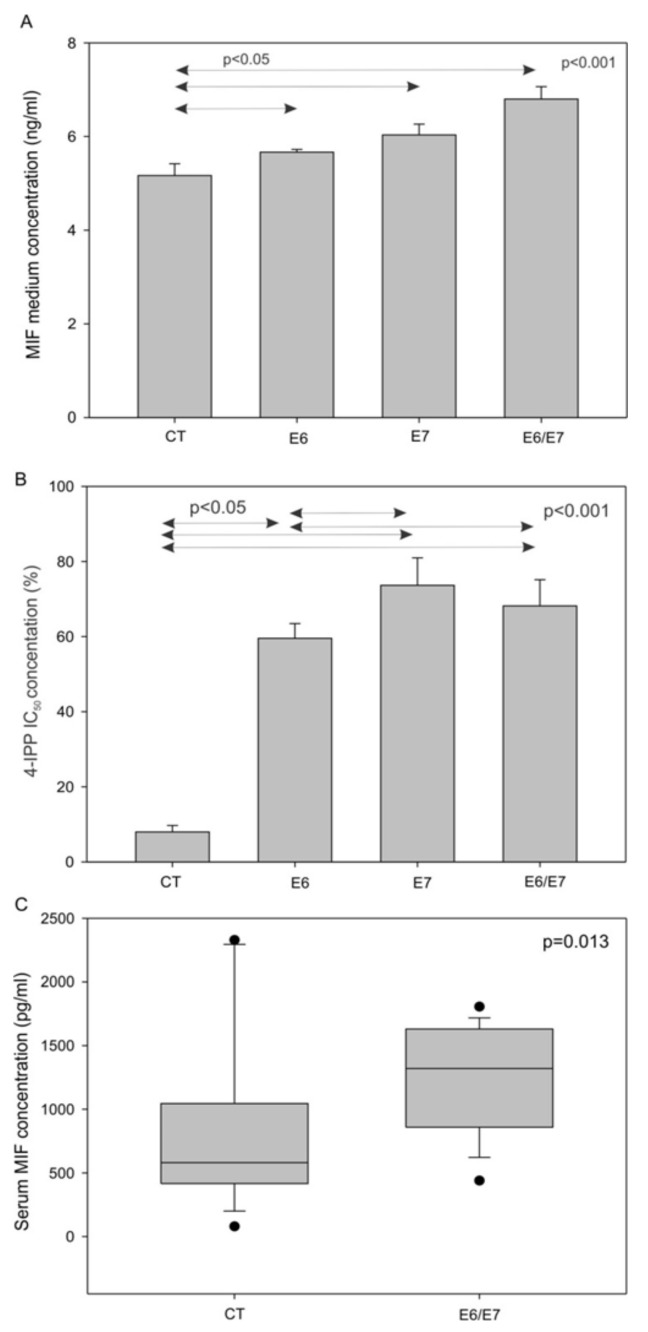
Macrophage migration inhibitory factor (MIF) secretion by murine cells in vitro and in vivo, and 4-IPP IC50 in murine cell lines. (**A**) Increase in the MIF concentration in the culture medium of SCCVII cells expressing HPV oncoproteins (*p* <0.001, one-way ANOVA). (**B**) Percentage of cell proliferation after exposure to 50 µM 4-iodo-6-phenylpyrimidine (4-IPP) in SCCVII CT, E6, E7 and E6/E7 cell lines (*p* <0.001, one-way ANOVA test). (**C**) Increase in the serum MIF levels of mice receiving SCCVII E6/E7 cells (n = 18), compared to the serum MIF levels of mice in the control group (CT) (n = 16) (*p* = 0.013, Mann–Whitney test).

**Figure 4 jcm-08-00075-f004:**
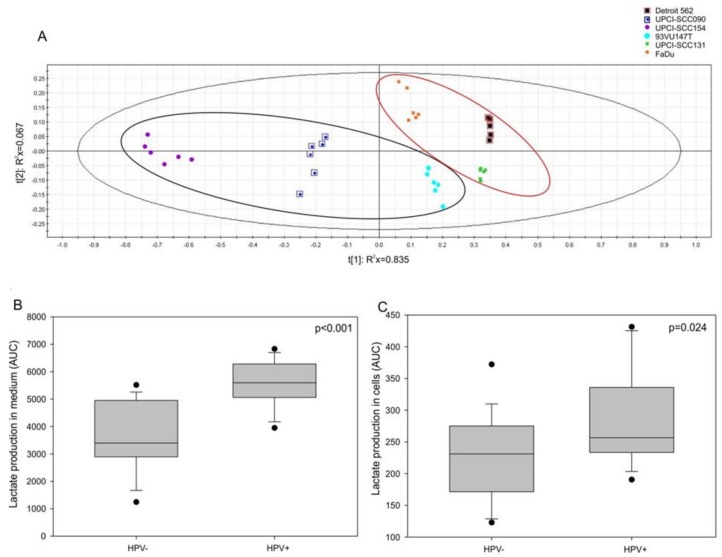
Lactate production in human cell lines. (**A**) Partial least-squares discriminant analysis (PLS-DA) for human papilloma virus (HPV)-positive and HPV-negative cell lines. Increase in lactate production in (**B**) the extracellular compartment (*p* <0.001, Mann-Whitney test) and in (**C**) the intracellular compartment (*p* = 0.024, Mann-Whitney test) of HPV-positive and HPV-negative cells.

**Figure 5 jcm-08-00075-f005:**
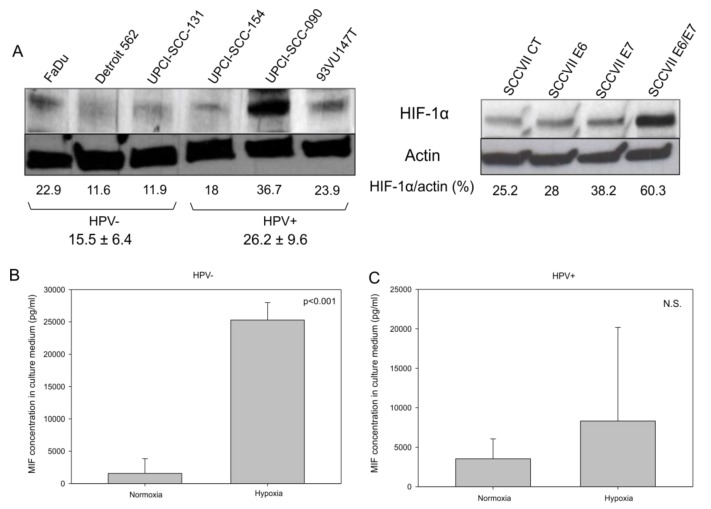
Hypoxia inducible factor 1α (HIF-1α) expression in human and murine cell lines. (**A**) Western blot analysis demonstrating the upregulation of HIF-1α in two human papilloma virus (HPV)-positive cell lines (UPCI-SCC-090 and 93VU147T) and in the murine SCCVII E6/E7 cell line. Concentration of macrophage migration inhibitory factor (MIF) in the culture medium of (**B**) human HPV-negative cell lines (*p* <0.001, Student’s *t*-test), and (**C**) in human positive cell lines (N.S., Student’s *t*-test) under normoxic and hypoxic conditions.

**Figure 6 jcm-08-00075-f006:**
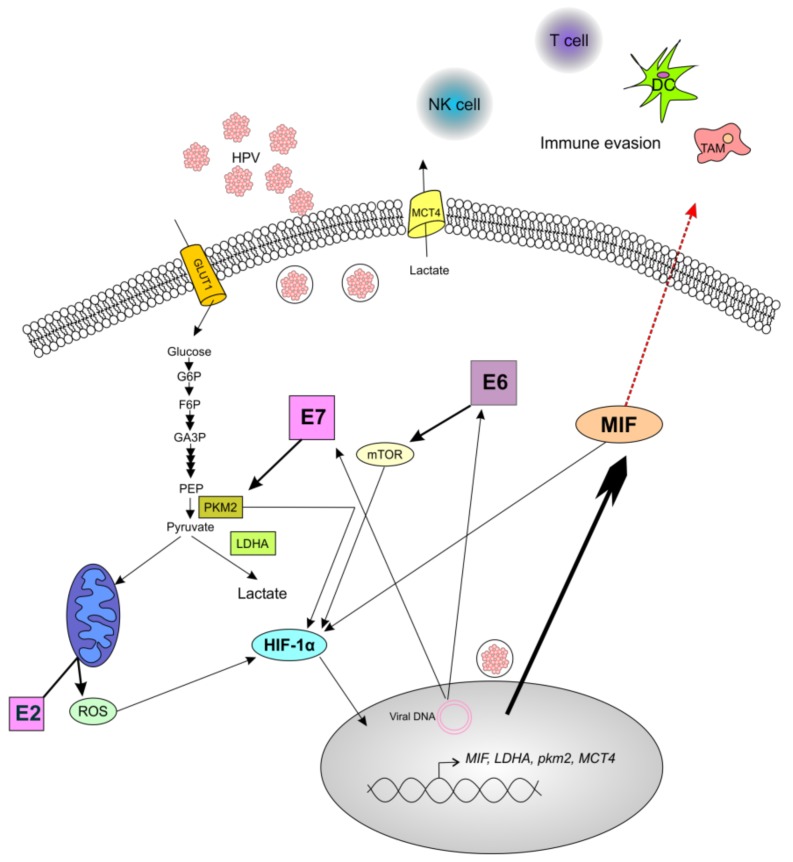
Schematic illustration of the role of human papilloma virus (HPV) oncoproteins in the production of macrophage migration inhibitory factor (MIF). The oncoprotein E6 promotes the activation of the mTOR signaling pathway, thus leading to the upregulation of hypoxia inducible factor 1α (HIF-1α), which enhances the production of MIF and increases the production of lactate. The oncoprotein E7 drives the accumulation of the dimeric form of pyruvate kinase M2 (PKM2), thus leading to the conversion of pyruvate to lactate through lactate dehydrogenase (LDHA). Moreover, PKM2 interacts with HIF-1α to amplify the expression of HIF-1α target genes, including MIF and LDHA. Additionally, the HPV E2 protein localizes to the mitochondrial membrane, leading to the increased production of reactive oxygen species (ROS), which increases the stability of HIF-1α. The enrichment of MIF in the extracellular environment could promote the tumor immune escape by the accumulation of pro-tumoral immune cells such as tumor-associated macrophages (TAM), and by the decrease in antitumoral cells such as cytotoxic T cells, Langerhans/dendritic cells (DC), and natural killer (NK) cells. Complete arrows indicate data from the literature, and the red dotted arrow indicates a hypothesis.

**Table 1 jcm-08-00075-t001:** Oropharyngeal cancer patients’ characteristics.

	Number of Cases
**Age (years)**	
Median	57
Range	44–90
**Gender**	
Male	31
Female	8
**Localization**	
Tonsils	13
Soft palate	11
Base of tongue	6
Posterior wall	3
Unknown	6
**Histological grade**	
Well differentiated	14
Moderately differentiated	12
Poorly differentiated	6
Unknown	7
**TNM stage**	
T1-2	19
T3-4	10
Unknown	10
**Risk factors**	
*Alcohol*	
Non-drinker	3
Drinker	24
Unknown	12
*Tobacco*	
Non-smoker	2
Smoker	25
Unknown	12
*HPV status*	
HPV−ve	21
HPV+ve/p16+	3
HPV+ve/p16−	10
HPV+ve/p16 unknown	5
**Recurrence**	
Yes	15
None	15
Unknown	9
**Clinical follow-up (months)**	
Median	33
Range	1–121

**Table 2 jcm-08-00075-t002:** Oral cavity cancer patients’ characteristics.

	Number of Cases
**Age (years)**	
Median	58
Range	23–87
**Gender**	
Male	97
Female	20
**Localization**	
Tongue	27
Floor of the mouth	35
Gingivae	12
Buccal mucosa	7
Retromolar trigone	3
Lips	2
Hard palate	1
Unknown	30
**Histological grade**	
Well differentiated	44
Moderately differentiated	53
Poorly differentiated	19
Unknown	1
**TNM stage**	
T1-2	79
T3-4	31
Unknown	7
**Risk factors**	
*Alcohol*	
Non-drinker	47
Drinker	49
Unknown	21
*Tobacco*	
Non-smoker	40
Smoker	57
Unknown	20
*HPV status*	
HPV−ve	65
HPV+ve/p16+	6
HPV+ve/p16-	42
HPV+ve/p16 unknown	4
**Recurrence**	
Yes	44
None	59
Unknown	14
**Clinical follow-up (months)**	
Median	32
Range	1–188

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
