# Peer review of "Involvement of HPV Infection in the Release of Macrophage Migration Inhibitory Factor in Head and Neck Squamous Cell Carcinoma"

_jcm, 2019, doi:10.3390/jcm8010075_

Round 1

Reviewer 1 Report

Kindt et.al studies the role of the MIF in the HPV positive and negative cell line in HNSCC. The manuscript is well written. Nonetheless, the study is interesting and provides some insights in the MIF role. The manuscript can be accepted after addressing the concern

1.       The expression of MIF was studied in the different grade as well as tumor sites. The authors can discuss the observation on this.

2.       It is advisable to check what will be the effect of MIF knockout in the cell line model. Whether the authors can recapitulate the same results they generated with MIF inhibitor

3.       In the manuscript method section LC-MS/MS analysis paragraph were mentioned. The results from this analysis were not descried in either result or discussion section.

4.       The raw data for the LC-MS/MS need to submit in public repository and it should made available to the community

5.       Figure 2 C MIF western for HPV positive and negative cell line there is no significant change

6.       Typo error C18 column line 314, 18 should be in subscript

Author Response

We would like to thank you for reviewing our paper and providing relevant comments that will help us to improve the quality of our publication. You will find below the corrections made in our article (in red) in response to your recommendations.

1.       The expression of MIF was studied in the different grade as well as tumor sites. The authors can discuss the observation on this.

We added a sentence in the discussion to say that there is no correlation between MIF expression and clinical characteristics.

2.       It is advisable to check what will be the effect of MIF knockout in the cell line model. Whether the authors can recapitulate the same results they generated with MIF inhibitor

Our results have demonstrated in vivo that the expression of oncoproteins E6 and E7 in SCCVII cell line lead to increased secretion of MIF. To further support this observation and reply to the reviewer, we have inserted the result of the MIF knockdown (KD) in SCCVII line where we showed that the MIF KD cells secrete lesser MIF than MIF sc cells (control) (Figure 2E).

3.       In the manuscript method section LC-MS/MS analysis paragraph were mentioned. The results from this analysis were not descried in either result or discussion section.

Our study is focused on MIF expression and secretion and LC-MS/MS analysis was used as a tool to explore this aim. A complete analysis of the results from the proteomic data are still under examination and will be the subject of another article.

4.       The raw data for the LC-MS/MS need to submit in public repository and it should made available to the community

As we explain before, the results from the proteomic analyze will be the subject of another article where the raw data from the LC-MS/MS will be submit in public repository. However, if it is required by the editor, we could send these raw data.

5.       Figure 2 C MIF western for HPV positive and negative cell line there is no significant change

Indeed, there is no statistical difference between MIF expression in HPV+ and HPV- cell lines but we could see that 2 of the HPV- cell lines express more MIF than HPV+ cell lines.

6.       Typo error C18 column line 314, 18 should be in subscript

As you requested, the typo was changed.

Reviewer 2 Report

1)      A more detail literature review is required. Following are some suggestions.

a.       approximately 600,000 new cases in 2012 [1]>> a more recent data will be useful. This data is almost 6 year old.

b.       associated with HPV-16 and HPV-18 types [2]>>Authors should also refer to (PMID: 25925419) which has detailed information about the involvement of HPV subtype with HNSCC.

c.       The impact of HPV infection remains controversial concerning the prognostic values of HNSCC.>>Recently researchers at UNC have shown the prognostic values of HPV. Authors should review their work. International Journal of Radiation Oncology• Biology• Physics, 2018, 100 (5), 1310–1311 and International Journal of Radiation Oncology• Biology• Physics 2018, 102 (5), 1605–1606

d.       HPV infection in oropharyngeal cancer is nearly 70% in North America>>ref?

e.       it is approximately 40% in Europe>>ref? Also, More studies on the percentage of HNSCC because of HPV are reported from different countries like Sweden: Int J Cancer. 2009;125(2):362-6; Canada: Cancer. 2010;116(11):2635-44; Australia: Br J Cancer. 2011;104(5):886-91

f.        The oncoproteins E6 and E7 play a crucial role in the neoplastic transformation of host cells by inactivating p53 and pRb, respectively>> reference required for this statement.

2)      In section 2.1 authors have shown that “Tissue MIF Expression is Decreased in Oral Cavity and Oropharyngeal Carcinomas Infected with HPV” while in section 2.2 it is shown that “MIF mRNA Synthesis and Protein Secretion is Increased in HPV Infected Human Head and Neck Cancer Cell Lines”. It seems contradictory. Authors have shown the increase in the MIF expression in most of their results. The logic of decrease in MIF expression and increase of MIF secretion is not clear. How decrease in MIF expression is leading to the increase secretion?

3)      The loading plot showed that only lactate come through the analysis (data not shown)>> There should not be any data not shown statement. Authors should include all relevant information (not suitable for main text) to supplementary material.

4)      under hypoxic conditions which stimulate HIF-1α expression and activity (data not shown)>> comment same as no. 3.

5)      “Moreover” word is used multiple times which his interfering with the flow of the reading. Authors should reduce the frequency of this word in the text.

6)      “In conclusion, we have established….. development of a protumoral microenvironment.”>> The conclusion statement is presented as the theory while it should be proposed as a hypothesis. Authors do not have sufficient data to say such a strong word. They need to confirm by multiple ways before they present conclusion in such strong words….. in the same context “section 5 Conclusions” should also be presented in a same way.

7)      Although authors have mentioned about the ethical clearance in Clinical Characteristics but ideally ethical statement should have a separate heading. Authors have not mentioned that all the patients have given written consent.

Minor points

1)      The current objective of our research>> The objective of our research

2)      HPV- and HPV+>> can be written as HPV-ve and HPV+ve for better clarity.

Author Response

We would like to thank you for reviewing our paper and providing relevant comments that will help us to improve the quality of our publication. You will find below the corrections made in our article (in red) in response to your recommendations.

1)      A more detail literature review is required. Following are some suggestions.

a.       approximately 600,000 new cases in 2012 [1]>> a more recent data will be useful. This data is almost 6 year old.

As you requested, we have changed the reference to a more recent one (Global Burden of Disease Cancer Collaboration et al, JAMA Oncol, 2017).

b.       associated with HPV-16 and HPV-18 types [2]>>Authors should also refer to (PMID: 25925419) which has detailed information about the involvement of HPV subtype with HNSCC.

We have added the reference in the text.

c.       The impact of HPV infection remains controversial concerning the prognostic values of HNSCC.>>Recently researchers at UNC have shown the prognostic values of HPV. Authors should review their work. International Journal of Radiation Oncology• Biology• Physics, 2018, 100 (5), 1310–1311 and International Journal of Radiation Oncology• Biology• Physics 2018, 102 (5), 1605–1606

We have included these two pertinent references in the text (line 42-44).

d.       HPV infection in oropharyngeal cancer is nearly 70% in North America>>ref?

We have added a reference (line 47).

e.       it is approximately 40% in Europe>>ref? Also, More studies on the percentage of HNSCC because of HPV are reported from different countries like Sweden: Int J Cancer. 2009;125(2):362-6; Canada: Cancer. 2010;116(11):2635-44; Australia: Br J Cancer. 2011;104(5):886-91

We have inserted the reference (lines 47 and 48). Indeed, this study of Anantharaman et al in 2017 analyze the prevalence of HPV infection in HNSCC notably in United States and in Europe including tumor samples from Germany, UK and Italy. To be more complete, we have also included the reference of Näsman et al from Sweden.

f.        The oncoproteins E6 and E7 play a crucial role in the neoplastic transformation of host cells by inactivating p53 and pRb, respectively>> reference required for this statement.

As you requested, we have added a reference at line 205.

2)      In section 2.1 authors have shown that “Tissue MIF Expression is Decreased in Oral Cavity and Oropharyngeal Carcinomas Infected with HPV” while in section 2.2 it is shown that “MIF mRNA Synthesis and Protein Secretion is Increased in HPV Infected Human Head and Neck Cancer Cell Lines”. It seems contradictory. Authors have shown the increase in the MIF expression in most of their results. The logic of decrease in MIF expression and increase of MIF secretion is not clear. How decrease in MIF expression is leading to the increase secretion?

Indeed, we showed that MIF expression increase in tumor cells of HPV- HNSCC patients but in vitro we demonstrated that MIF is more secreted by HPV+ cell lines. As explained in the Discussion part, line 257, we hypothesized that MIF expression decrease in HPV positive tissues because most of the MIF production is secreted probably by a mechanism of MIF transport still unknown in HPV+ cells.

3)      The loading plot showed that only lactate come through the analysis (data not shown)>> There should not be any data not shown statement. Authors should include all relevant information (not suitable for main text) to supplementary material.

Unfortunatly, we have no additional information to show due to data lost.

4)      under hypoxic conditions which stimulate HIF-1α expression and activity (data not shown)>> comment same as no. 3.

It is a mistake, the data is shown in the Figure 5, graph B.

5)      “Moreover” word is used multiple times which his interfering with the flow of the reading. Authors should reduce the frequency of this word in the text.

We have used synonym of moreover to replace this word in some part of the text.

6)      “In conclusion, we have established….. development of a protumoral microenvironment.”>> The conclusion statement is presented as the theory while it should be proposed as a hypothesis. Authors do not have sufficient data to say such a strong word. They need to confirm by multiple ways before they present conclusion in such strong words….. in the same context “section 5 Conclusions” should also be presented in a same way.

We have changed the sentences distinguishing conclusions and subsequent hypothesis.

7)      Although authors have mentioned about the ethical clearance in Clinical Characteristics but ideally ethical statement should have a separate heading. Authors have not mentioned that all the patients have given written consent.

We have completed the ethical section by adding the sentence “written informed consent was obtained from the patients”.

Minor points

1)      The current objective of our research>> The objective of our research

We have suppressed “current” in the introduction.

2)      HPV- and HPV+>> can be written as HPV-ve and HPV+ve for better clarity.

We added the -ve behind HPV- or HPV+.

Round 2

Reviewer 2 Report

Authors have substantially improved the manuscript by addressing all the raised questions. At this point, I do not have additional comments. 

Minor point:

1) county of Stockholm>>country should be Sweden instead of Stockholm.